# Towards a more Holistic Evaluation of Object-Centric Learning

## Abstract

Object-centric learning (OCL) methods were developed by taking inspiration from how humans perceive a scene. It is conjectured that they achieve compositional generalisation by decomposing the scene into objects, making the learned models robust to out-of-distribution (OOD) scenes. However, the recent OCL literature, by and large, evaluates the learned models only on the proxy task of object discovery, which gives no information about which object properties are actually encoded in the object-centric latent representation. Moreover, these models are not evaluated for the broader goals behind object-centric methods such as compositional generalisation, OOD performance, counterfactual reasoning, *etc*. Our work argues that the present evaluation protocols for OCL methods are significantly limited or not scalable. We propose using vision-language models (VLMs) for probing OCL methods on various visual question answering tasks. We are the first to evaluate OCL methods on multiple dimensions, ranging from counterfactual reasoning, OOD generalization, and robustness to natural adversarial examples. We also propose a new metric that unifies the evaluation of the 'what' and 'where' attributes, making the evaluation of OCL methods more holistic compared to existing metrics. Finally, we complement our analysis with a simple multi-feature reconstruction-based OCL method that outperforms the state of the art across several tasks.

## 1 Introduction

Object-centric learning (OCL) aims to decompose a scene into a set of latent representations. Instead of using a global encoding, object-centric representations help in tasks that require reasoning at an object level. OCL methods aim to enable vision systems to reason about a scene by decomposing it into its constituent objects, akin to how humans reason about a scene (Baillargeon et al., 1985; Spelke, 1990; Téglás et al., 2011). It is thought that reasoning about objects in a scene enables compositional or systematic generalisation (Greff et al., 2020; Wiedemer et al., 2024; Kapl et al., 2025), leading OCL methods to be more robust to out-of-distribution samples (Dittadi et al., 2021; Arefin et al., 2024) and enabling causal reasoning (Schölkopf et al., 2021; Mansouri et al., 2023). Among various approaches (Greff et al., 2019; Engelcke et al., 2020; Lin et al., 2020), slot-attention-based methods (Locatello et al., 2020) have gained popularity for their strong performance on real-world data (Everingham et al., 2010; Lin et al., 2014). Slot-attention methods find adoption in diverse areas, such as building world models, robotics, compositional generation, and visual navigation (Li et al., 2021; Huo et al., 2023; Wu et al., 2023a; Villar-Corrales & Behnke, 2025).

For OCL methods to be successfully applied to these diverse areas, they need to learn object representations that both capture the object properties ('what') and also the location of the object ('where'). However, existing evaluation schemes suffer from two key issues: *(1) Limited evaluation of broader properties.* Scaling existing schemes for evaluating the representation quality, such as linear probing, is infeasible due to the requirement of a large paired dataset for training and the limited capacity of linear probes. Moreover, these linear probes themselves do not generalise well beyond their training data, making them unable to evaluate broader goals behind OCL methods like out-of-distribution (OOD) generalisation, counterfactual reasoning, and compositional generalisation to new scenes. Mamaghan et al. (2025) proposed using transformer-based probes trained with visual question answering (VQA) datasets. However, their evaluation framework is also trained with small VQA datasets, making it hard to evaluate broader properties like counterfactual reasoning, OOD generalisation, and more. *(2) Disjoint evaluation metrics.* Separately evaluating the 'what' and 'where'

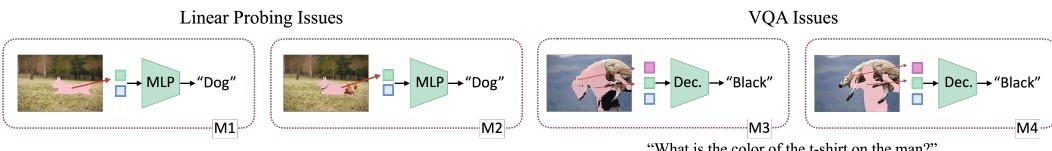

Figure 1: **Limitations of existing OCL evaluation schemes.** *(Left)* Using linear probing to evaluate OCL models M1 and M2 would result in assigning the same score for both models, but as seen, M1 localises the object much better than M2. We refer to this as *Type1* inconsistency. *(Right)* Using VQA as an evaluation metric is problematic, too, due to additional *Type2* inconsistencies: We cannot be sure which slot was responsible for answering the question. For example, when evaluating via VQA, model M4, which correctly answers the question using the correct slot, is given the same score as model M3, which uses the slot bound to sheep to answer the question, albeit correctly.

qualities leads to the following inconsistencies: (*Type1* inconsistencies) A model predicts properties well but fails to localise the object correctly. (*Type2* inconsistencies) Multiple slots may redundantly encode the same object, leading to representation fragmentation. Linear probing suffers from *Type1* inconsistencies, while VQA evaluation suffers from both *Type1* and *Type2* inconsistencies (see Fig. 1).

We propose to use large language models (LLMs) to evaluate the representational performance of object-centric learning methods. Specifically, we employ the visual instruction tuning method of Liu et al. (2023), which modifies an LLM into a vision-language model (VLM). We use object-centric models as the vision encoders, enabling us to evaluate OCL methods through visual question answering (VQA) via the VLM. By using a pre-trained LLM for reasoning, we can leverage the open-world capabilities of LLMs and assess OCL models efficiently on multiple VQA datasets and therefore more holistically across a wide range of dimensions, including counterfactual reasoning, out-of-distribution generalisation, *etc*., and thus tackle issue *1*. However, simply using VLMs for evaluation still suffers from *Type1* and *Type2* inconsistencies due to the disjoint evaluation of localisation and property prediction capabilities. To tackle this issue, we introduce a new metric, attribution-aware grounded accuracy (AwGA), which jointly considers both object localisation and property prediction capabilities when evaluating an OCL model and also tackles the representation fragmentation issue.

In summary, our work aims to build a holistic evaluation framework for object-centric learning methods. Our contributions are as follows: *(i)* We propose a general-purpose evaluation framework for object-centric methods based on visual instruction tuning (Liu et al., 2023), where the image encoders are OCL models. *(ii)* Our work is the first to benchmark several state-of-the-art (SOTA) OCL methods along diverse dimensions like compositional understanding, counterfactual reasoning, and out-of-distribution robustness. We show that OCL methods are comparable to foundational models such as DINOv2 (Oquab et al., 2024) on many of these settings. *(iii)* We propose a new metric, called attribution-aware grounded accuracy (AwGA). AwGA is the first metric that provides a holistic benchmark that jointly evaluates the 'where' and 'what' properties of OCL methods and tackles the issue of *Type1* and *Type2* inconsistencies. *(iv)* Based on our assessments, we propose a simple object-centric learning method named mFRESA, which uses multiple features in the reconstruction loss to learn better object-centric representations, outperforming SOTA OCL methods on many tasks.

## 2 RELATED WORK

**Object-centric learning (OCL)** seeks to learn a latent representation for each object in a scene *without supervision*, enabling more robust and compositional representations (Dittadi et al., 2021; Wiedemer et al., 2024) that can be applied to any scene in general. Early OCL works built on the VAE architecture (Burgess et al., 2019; Greff et al., 2019) but were limited by scalability and permutation variance. In order to overcome these issues, slot attention (SA; Locatello et al., 2020) introduced iterative clustering of features via attention, though initially had been restricted to synthetic data (Johnson et al., 2017; Groth et al., 2018; Karazija et al., 2021). Seitzer et al. (2023) scaled SA to real-world scenes (Everingham et al., 2010; Geiger et al., 2013; Lin et al., 2014) by reconstructing DINO features (Caron et al., 2020) instead of pixels. Modern approaches can broadly be classified based on their reconstruction loss: image-based methods (Jiang et al., 2023; Wu et al., 2023b; Singh et al., 2025; Akan & Yemez, 2025) rebuild pixels with strong decoders like StableDiffusion (Rombach

et al., 2022), whereas feature-based ones (Seitzer et al., 2023; Kim et al., 2024; Kakogeorgiou et al., 2024) reconstruct pre-trained features from DINO/DINOv2 (Caron et al., 2020; Oquab et al., 2024).

**Evaluation of OCL methods.** Object-centric learning methods can be considered a general class of representation learning methods, similar to foundational models like CLIP (Radford et al., 2021), DINO (Caron et al., 2020; Oquab et al., 2024), VQ-VAE (Van Den Oord et al., 2017), *etc*. A key distinction between OCL and these foundational methods is that OCL methods learn a unique latent representation per object in the scene. The unsupervised object discovery (UOD) task is the most popular way of evaluating OCL methods. However, as Rubinstein et al. (2025) pointed out, the UOD task is a poor proxy, as it does not assess several important properties, such as compositional generation, counterfactual reasoning, and OOD generalisation. Linear probing has been used for evaluating the representation quality of slots (Locatello et al., 2020; Jiang et al., 2023; Singh et al., 2025). However, using linear probing for evaluating multiple dimensions is not optimal because: *(i)* It is expensive to scale as testing different dimensions of OCL representations would require repeatedly retraining the linear probe for different datasets and benchmarks. *(ii)* Testing some properties, *e.g.*, counterfactual reasoning, with linear probing is non-trivial as it requires a discrete output space, which can grow easily very large, making optimization difficult. Alternatively, grouping the output space into classes is inherently ambiguous and would result in a coarse evaluation. To tackle these limitations, Mamaghan et al. (2025) proposed using VQA to evaluate OCL methods. However, unlike our evaluation framework, their approach cannot evaluate slot-attention-based methods across multiple dimensions as it requires expensive re-training of the transformer model from scratch repeatedly on each new dataset. For example, Kapl et al. (2025) adapted it to evaluate compositional generalization by retraining on a custom compositional dataset. In addition, both rely solely on accuracy as an evaluation metric, which makes them susceptible to *Type2* inconsistencies. Thus, a need exists for a single framework which can evaluate a wide range of properties and accounts for *Type1* and *Type2* inconsistencies. In our work, we aim to bridge this gap.

**Going beyond linear probing, using vision-language models (VLM) as evaluators.** Linear probing and end-to-end finetuning have been popular ways of evaluating representation learning methods, including OCL methods (He et al., 2022; Seitzer et al., 2023; Jiang et al., 2023; Oquab et al., 2024). However, recently Tong et al. (2024) questioned their use, stating that these do not reflect diverse perception challenges of the real world. Cambria-1 (Tong et al., 2024) uses a visual instruction tuning setup (Liu et al., 2023) for evaluating the performance of several vision encoder models. They show that using visual instruction tuning allows the utilization of several high-quality datasets and benchmarks (Goyal et al., 2017; Hudson & Manning, 2019; Singh et al., 2019; Yu et al., 2023; Fu et al., 2024) to evaluate diverse properties of vision encoders, going far beyond simple linear probing. Inspired by this, we employ visual instruction tuning for evaluating OCL methods as vision encoders in the vision language model. However, directly applying LLaVA-style training (Liu et al., 2023) does not assess if the slots have a direct correspondence to objects in a scene, *i.e.* a slot binds to single object and only encodes its properties. To this end, we propose an attribution-aware grounded accuracy (AwGA) metric, which takes into account *Type1* and *Type2* inconsistencies when evaluating OCL methods.

## 3 VLMs as Evaluators of Object-Centric Representations

### 3.1 Preliminaries

**Slot Attention** (**SA**; Locatello et al., 2020) is an iterative refinement framework that decomposes an image into a set of object-centric representations called slots. These slots are learned by extracting the input image's feature map $\mathbf{H}$ using an encoder network. After this, the slot-attention module iteratively groups this feature map into a set of $k$ slot vectors $\mathbf{S} = \{\mathbf{s}_1, ..., \mathbf{s}_k\}$. At each iteration $t$, the slot representation $\mathbf{S}^t$ is updated using the dot-product attention (Vaswani et al., 2017) between the previous slot representation $\mathbf{S}^{t-1}$ and the input feature vectors $\mathbf{H}$. Unlike the traditional attention mechanism, which performs a softmax operation over keys, here, the softmax operation is over slots. This creates a competition between the slots to explain part of the input image, making slots bind to objects. For further details, see (Locatello et al., 2020). Typically, SA methods use 7 slots for real-world images (for example from the COCO dataset; Lin et al., 2014), as this setting has been found to yield the strongest performance on both object discovery and certain representation metrics.

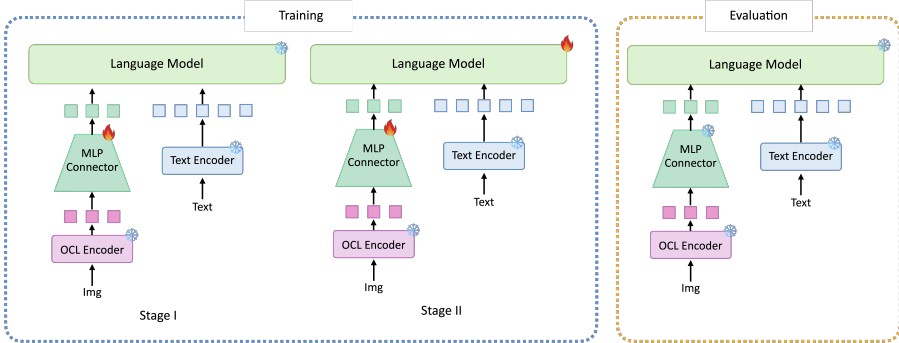

Figure 2: **Training setup.** Our training is akin to LLaVA (Liu et al., 2023). In Stage I, only the MLP connector is trained on the pre-training dataset. This stage aligns the slot embeddings to the space of the language model. In Stage II, the MLP network and the language model are trained on the instruction tuning dataset from LLaVA. This stage tunes the language model to follow instructions and perform tasks based on slots as visual tokens. Evaluation is performed on various VQA benchmarks, where the text is encoded via a text encoder and images are encoded using a slot-attention model.

### 3.2 VLMS AS EVALUATORS

As motivated above, we here propose to use VLMs as evaluation engines for OCL methods. We can formalise this setup as a function composition $f(g(\mathbf{S}))$, where $f$ denotes an LLM and $g$ denotes a connector network (*e.g.*, 2-layer MLP) that connects the LLM to the slot-representation $\mathbf{S}$ to be evaluated and needs to be trained. This VLM evaluation protocol can be seen as a more powerful and capable generalization of previous evaluation protocols; linear probing ($f = \mathbf{I}$, $g = $ 1-layer MLP) and transformer-based probes (Mamaghan et al., 2025) ($f = \mathbf{I}$, $g = n$-layer transformer), where $\mathbf{I}$ denotes the identity function, are special cases.

Our evaluation protocol for assessing OCL methods utilises the visual question-answering task. Inspired by LLaVA (Liu et al., 2023), we follow their architecture and training protocol for learning a vision-language model, where we use object-centric models as vision encoders. The training process (shown in Fig. 2) has two stages: *(i)* In Stage I, the alignment stage, the slot embeddings are projected by a connector network to align with text embeddings so they are in the same space. Only the connector network is trained for one epoch on the LLaVA 558K pre-training dataset in this stage. *(ii)* In Stage II, the instruction tuning phase, the model is trained with the LLaVA 665K instruction tuning dataset, which comprises multimodal samples created via GPT-4's responses (Achiam et al., 2023) to images in the COCO dataset. For more details, see (Liu et al., 2023). The instruction tuning phase helps the model follow instructions better and improves the VLM's ability to respond accurately and effectively to user prompts. In this stage, the MLP connector network and the language model are trained simultaneously for one epoch. The full training setup is illustrated in Fig. 2.

## 4 ANALYSIS OF OCL METHODS

In Sec. 4.1, we first benchmark the representational power of embeddings learned by object-centric methods using tasks such as counterfactual reasoning, out-of-distribution generalisation, *etc*. Sec. 4.2 shows that the prevalent evaluation scheme using object discovery is a poor proxy for the representation capabilities of OCL methods. Further, Sec. 4.3 introduces our new metric AwGA, addressing both *Type1* and *Type2* errors, and used for OCL benchmarking in Sec. 4.4. We begin by explaining the experimental setup and our improved baseline mFRESA.

**Baselines.** The original goal of object-centric learning (OCL) has been to obtain object-centric representations in an *unsupervised* manner; we thus focus our evaluation on unsupervised OCL methods. We take state-of-the-art baselines for real-world datasets (*e.g.*, COCO; Lin et al., 2014), including SPOT (Kakogeorgiou et al., 2024), SlotDiffusion (Wu et al., 2023b), StableLSD (Jiang et al., 2023), FT-DINOSAUR (Didolkar et al., 2025b), and DINOSAUR (Seitzer et al., 2023). Whenever available, we use the authors' released checkpoints; StableLSD and DINOSAUR are re-trained from scratch using their official scripts. We also include a DINOSAURv2 baseline, which replaces

Table 1: **VQA comparison of OCL and foundational models.** Blind-VLM denotes a variant of the VLM where the vision tokens are replaced with random noise. This serves as a lower bound, helping to disentangle the contributions of the LLM and vision encoder models, thereby providing more interpretability to the evaluation. DINOv2 serves as a reference upper bound for self-supervised representation learning against slot-attention models using feature reconstruction (*2^{nd} group*) and image reconstruction (*3^{rd} group*). We highlight the **best** and second best model among SA methods. We report the accuracy (in %, ↑) on the visual question answering task. For MME, we report results only on perception tasks (↑, maximum achievable score 2000). Additional details about the dataset and metrics are provided in Sec. C. DINOv2 outperforms OCL models across vision-centric VQA tasks. However, slot-attention methods—especially mFRESA—perform competitively despite using only 7 vision tokens (*vs.* 196 for DINOv2).

| LLM | Phi2 | | | | | Qwen2-7B | | | | |
|---|---|---|---|---|---|---|---|---|---|---|
| Dataset | GQA | POPE | MME | MMVET | VQAv2 | GQA | POPE | MME | MMVET | VQAv2 |
| Blind-VLM | 38.20 | 64.92 | 667.38 | 13.2 | 44.71 | 40.32 | 65.12 | 686.29 | 15.0 | 45.76 |
| DINOv2 | 57.77 | 82.01 | 1279.21 | 22.6 | 71.15 | 61.83 | 83.31 | 1388.19 | 23.6 | 74.86 |
| DINOSAUR | 49.71 | 78.72 | 1047.34 | 17.5 | 58.41 | 53.63 | 79.54 | 1178.87 | 15.9 | 61.98 |
| DINOSAURv2 | 53.23 | 81.76 | 1122.73 | **18.9** | 63.84 | 56.32 | 82.49 | 1224.84 | 17.7 | 66.23 |
| VQDINO$_{MLP}$ | 52.69 | 81.48 | 1167.08 | 18.8 | 63.22 | 56.29 | 82.36 | 1229.43 | 18.8 | 67.28 |
| FT-DINOSAUR | 52.22 | 81.45 | 1004.94 | 15.1 | 60.97 | 56.15 | 81.85 | 1242.25 | 17.2 | 66.09 |
| SPOT | 51.06 | 79.74 | 1069.80 | 17.4 | 60.94 | 54.94 | 80.24 | 1169.11 | 17.8 | 65.37 |
| Slot Diffusion | 50.00 | 79.77 | 1090.10 | 18.5 | 59.65 | 53.93 | 79.91 | 1171.83 | 18.9 | 63.47 |
| StableLSD | 51.45 | 81.51 | 1129.08 | 17.8 | 62.06 | 55.96 | 81.54 | 1239.48 | 18.6 | 66.67 |
| mFRESA *(ours)* | **53.90** | **82.12** | **1187.05** | 18.5 | **65.58** | **58.28** | **82.74** | **1283.48** | **19.3** | **69.93** |

the original DINO (Caron et al., 2021) backbone with DINOv2 (Oquab et al., 2024), and a vector quantized version of DINO (VQDINO$_{MLP}$) introduced by Zhao et al. (2025).

**Improved baseline.** Existing OCL methods often use either feature reconstruction (Seitzer et al., 2023; Kakogeorgiou et al., 2024) or image reconstruction (Jiang et al., 2023; Wu et al., 2023b) as their target. We propose a new simple OCL baseline (mFRESA), which combines multiple reconstruction targets such as pixels, features, and additionally HOG features. In particular, mFRESA is based on StableLSD (Jiang et al., 2023) with two additional decoders: *(1)* a feature decoder that reconstructs DINOv2 features akin to (Seitzer et al., 2023); *(2)* a HOG decoder based on a simple three-layer MLP that reconstructs gradient-based histograms computed over local patches (Dalal & Triggs, 2005). Below we show that reconstructing multiple features helps mFRESA learn more robust features, which encode more object information in the latents and also localise better. More details about the loss function, network diagram, and training for mFRESA can be found in Sec. A.

**Training details for VLM-based evaluation of OCL methods.** We use Phi2 (Javaheripi et al., 2023) and Qwen2-7B (Yang et al., 2024) as language models for VLM training, connected via a 2-layer MLP with GeLU activations (Hendrycks & Gimpel, 2016). Pre-training is performed with batch size 256 and learning rate 1e-3, followed by finetuning with batch size 128 and learning rate 2e-5, using AdamW (Loshchilov & Hutter, 2017) throughout. The maximum sequence length is set to 2048 tokens, and we adopt the official LLaVA dataset (Liu et al., 2023) for both pre-training and finetuning. During evaluation, we follow LLaVA and use greedy decoding (temperature 0, beams 1). All training and architectural settings are held fixed across LLMs and vision encoders, ensuring that the only variability comes from the vision encoder (*i.e.*, slot attention) module. VLM training is performed on 8×A100 GPUs (80 GB each). Training time varies with the OCL encoder, with the longest runs occurring for mFRESA and StableLSD. For these models, pre-training requires approximately 4 hours, and fine-tuning takes roughly 20–24 hours.

## 4.1 EVALUATING THE REPRESENTATION QUALITY OF OCL METHODS

**Standard perception evaluation.** Using VLMs as evaluators of vision encoders enables benchmarking across a wide range of VQA tasks. Since our goal is to assess the representation quality of vision tokens, we focus on image-centric datasets such as GQA (Hudson & Manning, 2019), VQAv2 (Goyal et al., 2017), MME (Fu et al., 2024), and MM-Vet (Yu et al., 2023). To evaluate whether OCL methods mitigate object hallucination—a common issue in large language models—we additionally use the POPE benchmark (Li et al., 2023), which probes object presence via Boolean questions. For

MME, we report only perception tasks, as these are most relevant to our setting. Additionally, to isolate the contribution of the vision tokens and disentangle it from the LLM's prior knowledge, we also report results for a VLM, in which all vision tokens are replaced with random noise (*Blind-VLM*). This helps to quantify the performance gain resulting from the representational power of the vision encoders.

As shown in Table 1, despite using far fewer visual tokens (7 *vs*. 196 in DINOv2), OCL methods perform rather competitively with DINOv2, a state-of-the-art self-supervised vision encoder. Interestingly, FT-DINOSAUR, the leading OCL model for object discovery, underperforms DINOSAURv2 on nearly all datasets, highlighting that object discovery metrics alone do not fully capture the quality of the slot representation. By contrast, our improved baseline mFRESA outperforms all OCL methods across most benchmarks (except MM-Vet), suggesting that incorporating multiple decoders can yield stronger object-centric representations.

> *Takeaway 1.* While OCL methods build on pre-trained feature encoders like DINOv2, their slot representations still lag behind the feature encoder itself on perception tasks, suggesting that object-centric representations are currently not as effective for general visual perception tasks.

**Robust perception evaluation.** Given that OCL methods underperform in general perception tasks, we ask if OCL methods hold value on tasks that they are conjectured to work well on (Greff et al., 2020; Wiedemer et al., 2024; Kapl et al., 2025). Using VLMs as evaluation engines allows us to repurpose diverse benchmark datasets that test properties such as compositional learning and out-of-distribution (OOD) generalisation. This enables the evaluation of an often overlooked but critical aspect of OCL methods. Results are shown in Table 2.

*Positives.* On VQA tasks with OOD images from the OODCV dataset (Tu et al., 2024), containing images with unusual textures or backgrounds rarely seen in daily life, FT-DINOSAUR and DINOSAUR perform the best among OCL methods. Moreover, these methods are comparable to DINOv2 despite using far fewer tokens, indicating that OOD generalisation benefits from object-centric representations. We find that on counterfactual question answering, particularly direct numeric queries such as "How many X would there be if two X were added or removed?", OCL methods are competitive with or better than DINOv2. Interestingly, on Boolean counterfactual questions, the non-visual model (*Blind-VLM*) performs best. We believe this is because Boolean questions in CVQA are structurally simple ("Would X still be true if Y changed?") and can often be answered using linguistic priors alone. Adding vision features introduces visual signals, which can interfere in these simple cases. Finally, we would like to emphasize that correctly answering counterfactual questions does not necessarily imply a thorough understanding of the underlying causal mechanisms.

*Negatives.* For benchmarking compositional reasoning, we evaluate on the SugarCrepe dataset (Hsieh et al., 2023). The task is to pick the correct caption for a given image when provided with a correct caption and a hard-negative caption (attribute swaps, object additions, or replacements) generated by an LLM (Achiam et al., 2023). We find that OCL methods lag behind DINOv2. Although mFRESA narrows the gap, there is little evidence to suggest that explicit object representations improve compositional reasoning. We also benchmark whether reasoning about objects can improve robustness against naturally adversarial examples. We use the NaturalBench (Li et al., 2024) dataset, which provides a pair of questions for two images in a set. The sets are designed in a way that a blind model cannot succeed, *i.e.*, giving the same answer regardless of the image. Solving NaturalBench requires a model to possess object recognition, attribute binding, and relation understanding skills. We again see a large gap between OCL methods and the DINOv2 model, showcasing the need for designing better OCL methods that capture the object properties more comprehensively.

> *Takeaway 2.* Benchmarking OCL methods on tasks such as counterfactual reasoning, OOD generalisation, and compositional reasoning is essential to assess whether they are closing the gap with self-supervised representation learning methods like DINOv2.

## 4.2 ARE OBJECT DISCOVERY AND REPRESENTATIONAL POWER CORRELATED?

Unsupervised object discovery (UOD) metrics such as mean best overlap (mBO; Pont-Tuset et al., 2016) and mean intersection over union (IoU) are widely used to evaluate slot-attention methods.

Table 2: **Robustness of OCL methods.** Evaluation on tasks beyond object discovery such as OOD generalisation, compositional understanding, counterfactual reasoning, *etc*. (accuracy in %, ↑). The Blind-VLM serves as a lower bound while DINOv2 serves as a reference upper bound for performance on these tasks. We highlight the **best** and second best model among SA methods. The datasets evaluate the following properties: CVQA (Zhang et al., 2024) – counterfactual reasoning, OODCV (Tu et al., 2024) – OOD generalisation, NeuralBench (Zhang et al., 2024) – robustness to natural adversarial examples, SugarCrepe (Hsieh et al., 2023) – vision-language compositionality.

| LLM | Phi2 | | | | | Qwen2-7B | | | | |
| --- | --- | --- | --- | --- | --- | --- | --- | --- | --- | --- |
| Dataset | CVQA | | OODCV | N. Bench | SugarC. | CVQA | | OODCV | N. Bench | SugarC. |
| | Direct | Boolean | | | | Direct | Boolean | | | |
| Blind-VLM | 28.00 | 71.41 | 50.98 | 0.42 | 49.42 | 32.69 | 65.57 | 51.01 | 0.52 | 53.29 |
| DINOv2 | 36.96 | 63.72 | 58.00 | 8.42 | 82.05 | 45.74 | 53.54 | 58.36 | 9.89 | 88.06 |
| DINOSAUR | 35.74 | 69.29 | 51.97 | 1.89 | 67.85 | 41.13 | 63.72 | 52.52 | 3.95 | 72.45 |
| DINOSAURv2 | 34.52 | 65.75 | 53.90 | 3.37 | 75.98 | 42.09 | **64.07** | 56.66 | 6.16 | 78.18 |
| VQDINO$_{MLP}$ | 35.13 | 66.37 | 52.03 | 3.89 | 73.05 | **42.34** | 55.66 | 53.18 | 6.07 | 80.16 |
| FT-DINOSAUR | **39.13** | 68.85 | 55.18 | 2.89 | 70.94 | 42.00 | 57.17 | 53.28 | 5.42 | 81.24 |
| SPOT | 36.35 | 69.47 | 53.34 | 2.42 | 71.65 | 41.83 | 57.61 | 54.07 | 3.68 | 74.08 |
| Slot Diffusion | 33.83 | 68.23 | 51.34 | 2.21 | 70.39 | 39.39 | 59.56 | 52.56 | 3.74 | 74.53 |
| StableLSD | 38.26 | **70.44** | 52.89 | 3.00 | 72.92 | 41.04 | 62.39 | 55.08 | 5.21 | 78.98 |
| mFRESA *(ours)* | 38.09 | 66.64 | **55.57** | **4.21** | **77.27** | 41.39 | 60.44 | **57.31** | **6.84** | **83.17** |

Table 3: **Object discovery (OD) and representational quality are uncorrelated.** FT-DINOSAUR scores highest on OD metrics (mBO$_i$, mIoU) but underperforms on various VQA tasks (all in %, ↑). All methods use DINOv2 as the backbone. The Spearman's rank correlation between accuracy and mIoU for these models is $-0.2$, indicating poor correlation. All experiments use the Phi2 LLM.

| Dataset | VQAv2 | Nat. Bench | Sugar Crepe | COCO | |
| --- | --- | --- | --- | --- | --- |
| Metric | | accuracy | | mIoU | mBO$_i$ |
| DINOSAURv2 | 63.84 | 3.37 | 75.98 | 27.25 | 28.42 |
| FT-DINOSAUR | 60.97 | 2.89 | 70.94 | **34.52** | **36.08** |
| StableLSD | 62.06 | 3.00 | 72.92 | 24.52 | 25.72 |
| mFRESA *(ours)* | **65.75** | **4.11** | **77.17** | 30.60 | 32.17 |

Yet, it remains unclear whether higher UOD scores entail that the model also captures the object properties (colours, shape, *etc*.) better. In Table 3, we compare several OCL methods across UOD metrics, general visual question answering, adversarial robustness, and compositional reasoning.

We find that UOD metrics (mIoU and mBO) poorly correlate with slot representation quality. For instance, FT-DINOSAUR, the SOTA OCL model for the object discovery task, performs worse than the DINOSAURv2 model on general VQA tasks and robustness assessments (compositional reasoning and natural adversarial robustness). We attribute this loss in performance to FT-DINOSAUR finetuning the DINOv2 encoder, whereas other models keep it frozen. Finetuning on a small dataset like COCO likely reduces generalisation (Mukhoti et al., 2024), weakening the learned slot representations.

> *Takeaway 3.* Object discovery metrics are not highly correlated with the quality of object representations learned by the slots. This indicates a need for newer metrics that evaluate both localisation *and* representation abilities of OCL methods.

## 4.3 AwGA metric – Unifying 'what' and 'where'

As just shown, object discovery metrics correlate poorly with the representational quality of the slots. However, using only a downstream task metric like accuracy on VQA tasks does not directly evaluate how well object representations are localised. Fig. 3 shows how presently the accuracy and mIoU metrics are evaluated in a disjoint manner. Evaluating models with disjoint metrics leads to *Type1* and *Type2* inconsistencies as explained above (Fig. 1).

Figure 3: **Metrics for evaluation of OCL methods.** An overview of various metrics, which can be used to evaluate OCL methods. We propose the attribution-aware grounded accuracy (AwGA), which takes into account *Type1* and *Type2* inconsistencies when evaluating the model.

Table 4: **Performance comparison of different models using G-Acc and AwGA metrics** (all in %, ↑). mFRESA outperforms all OCL methods on the Enhanced Grounded GQA dataset.

| LLM | Phi2 | | | | Qwen2-7B | | |
|---|---|---|---|---|---|---|---|
| Metric | mIoU | Acc. | G-Acc. | AwGA | Acc. | G-Acc. | AwGA |
| DINOSAUR | 50.52 | 60.13 | 30.80 | 16.08 | 64.05 | 32.71 | 16.77 |
| DINOSAURv2 | 47.99 | 66.27 | 32.40 | 18.10 | 68.54 | 33.18 | 18.10 |
| VQDINO$_{\text{MLP}}$ | 48.50 | 65.55 | 32.27 | 18.28 | 68.18 | 34.13 | 19.03 |
| FT-DINOSAUR | 59.09 | 61.05 | 33.94 | 19.49 | 65.21 | 36.28 | 20.18 |
| SPOT | 53.76 | 64.08 | 38.45 | 20.83 | 68.32 | 41.45 | 22.40 |
| Slot Diffusion | 54.91 | 61.54 | 34.39 | 16.45 | 65.75 | 36.91 | 18.24 |
| StableLSD | 47.94 | 64.64 | 31.53 | 19.46 | 69.02 | 33.84 | 20.56 |
| mFRESA *(ours)* | 56.92 | 67.58 | 39.20 | 22.44 | 71.41 | 41.33 | 22.43 |

A way to account for *Type1* inconsistencies when evaluating different models is to use the grounded accuracy (G-Acc; Hudson & Manning, 2019), which is defined as

$$\text{G-Acc} = \frac{1}{N} \sum_{i=1}^{N} \mathbb{1}(\hat{y} = y) \ \text{mIoU}(\mathcal{A}_{\text{pred}}, \mathcal{G}_{\text{GT}}). \tag{1}$$

Here, $\mathcal{A}_{\text{pred}}$ and $\mathcal{G}_{\text{GT}}$ denote the mask predicted from the slots and the ground-truth (GT) grounding masks. $y$ and $\hat{y}$ denote the GT and predicted label; $\mathbb{1}$ is the indicator function. The grounding masks are composed of the masks of all objects required for answering a question. G-Acc is a weighted accuracy metric, where the weight for each correct answer equals the mIoU overlap between the predicted and the ground-truth masks. G-Acc has issues in our context, however, as it does not consider which slot was used to answer the question. Specifically, G-Acc does not penalise a model when it distributes the representation of an object across multiple slots; we call this *Type2* inconsistency. In order to correctly evaluate a model, we propose an attribution-aware grounded accuracy metric (AwGA), which penalises a model for committing both *Type1* and *Type2* inconsistencies. The AwGA metric first computes the attribution map (Simonyan et al., 2013) of each slot for answering the question. We then select slots with the $K$-highest attributions and compute the grounded accuracy. $K$ is *not* a hyperparameter but is set for each question equal to the number of objects in the grounding mask. This way, the mIoU is only computed for the slots that are responsible for answering the question. AwGA is formally written as

$$\text{AwGA} = \frac{1}{N} \sum_{i=1}^{N} \mathbb{1}(\hat{y} = y) \ \text{mIoU}(\text{TopK}(\mathcal{A}_{\text{pred}}), \mathcal{G}_{\text{GT}}). \tag{2}$$

For computing each attribution, we simply use the value of the gradient of each slot with respect to the loss function (Simonyan et al., 2013; Springenberg et al., 2015). In particular, we compute the sensitivity ($\frac{\partial y}{\partial \mathbf{s}_i}$) of the output $y = f(g(\mathbf{S}))$ with respect to each input feature $\mathbf{s}_i$. We also experimented with other attribution methods, such as integrated gradients (Sundararajan et al., 2017), but found the AwGA metrics to be robust to the choice of the attribution method (see Table 5).

### 4.4 AwGA-based evaluation

To assess OCL methods with our proposed AwGA metric, we use the validation set of GQA (Hudson & Manning, 2019), a large-scale VQA dataset with grounding boxes for each question. To better align with our evaluation, we enhance GQA by converting bounding box annotations into masks using SAM2 (Ravi et al., 2025), with boxes as prompts. To ensure that grounded objects are salient,

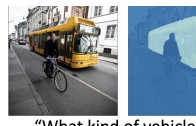 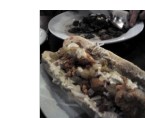 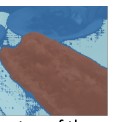 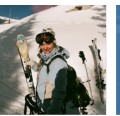 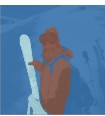

"What is this bird called?"  "What kind of vehicle is yellow?"  "Is the bread on top of the table"  "What is the woman holding?"

Figure 4: **Enhanced grounded GQA dataset.** Our AwGA-based evaluation uses this enhanced version, which contains both the input image and grounding masks, highlighting the objects necessary to answer the question.

we filter out images with more than seven boxes or those covering less than 10% of the image area. Examples are shown in Fig. 4 and more details provided in Sec. D.

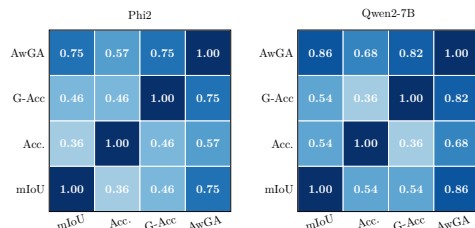

Figure 5: **Spearman's rank correlation between different metrics.** Our AwGA metric has a strong rank correlation to traditional mIoU and accuracy metrics.

Table 5: **Importance of attribution method for AwGA metric.** Comparison between using gradient and integrated gradients when calculating AwGA.

| | Attr. Type | |
|---|---|---|
| | **Grad.** | **Int. Grad.** |
| DINOSAURv2 | 18.10 | 17.75 |
| FT-DINOSAUR | 20.83 | 20.89 |
| StableLSD | 19.46 | 19.54 |
| mFRESA (ours) | **22.44** | **22.11** |

We report accuracy, mIoU, G-Acc, and our proposed AwGA metric in Table 4. The mIoU measures the overlap between predicted and ground-truth masks, but alone it cannot capture how well slots encode object properties. G-Acc penalises poor localisation but overlooks fragmented slot representations (*Type2* inconsistencies). By contrast, AwGA explicitly accounts for both localisation and representation quality, making it a more *holistic* metric.

Interestingly, models with top object discovery or accuracy scores are not always SOTA under G-Acc or AwGA, underscoring the pitfalls of disjoint evaluations. Moreover, AwGA shows strong Spearman rank correlations (Fig. 5.5) with *(1)* representation quality (Acc), and *(2)* localisation (mIoU), validating its role as a unified metric. Importantly, AwGA correlates more strongly with both accuracy and mIoU than accuracy and mIoU do with each other. mFRESA outperforms existing approaches on Acc, G-Acc, and AwGA, highlighting its stronger object-centric representations.

> *Takeaway 4.* Using only object discovery or accuracy metrics is an incomplete way of evaluating OCL methods. The proposed AwGA metric penalises methods with *Type1* and *Type2* inconsistencies, making it an important addition to the library of metrics for evaluating OCL methods.

**Robustness of AwGA to LLM and connector choice.** We next show that the AwGA metric is robust to both LLM and connector choices. To evaluate this, we compute the Spearman rank correlation of AwGA scores between Phi-2 and Qwen2-7B on the enhanced grounded GQA dataset. Then, using Phi-2 as the LLM, we evaluate three connector variants, 1-layer MLP (MLP1×), 2-layer MLP (MLP2×), and Q-Former, and again report Spearman correlations. As seen in Fig. 6, the correlations remain consistently high, showing that AwGA rankings are stable across LLMs and connectors.

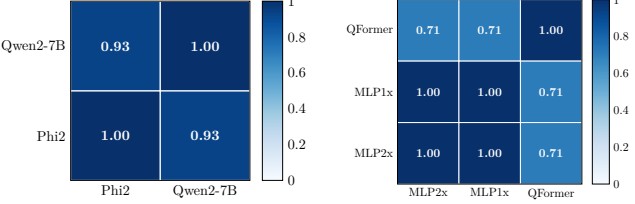

Figure 6: **Robustness of AwGA.** Spearman's rank correlation for the AwGA metrics for different LLM and connectors designs. AwGA remains stable across different LLMs and connector architectures.

Table 6: **Importance of HOG and feature decoders.** Both HOG and (DINOv2) feature decoders improve the performance, indicating their importance. All experiments (in %, ↑) use a Phi-2 LLM.

| Model | GQA | OOD | Sugar Crepe | VQAv2 | mIoU | AwGA |
|-------|-----|-----|-------------|-------|------|------|
| Image Only (StableLSD) | 51.45 | 52.89 | 72.92 | 62.06 | 47.94 | 19.46 |
| Image + Feat. Dec. | 52.21 | 54.20 | 75.92 | 61.40 | 55.76 | 22.03 |
| Image + Feat. Dec. + HOG Dec. | **53.90** | **55.27** | **77.27** | **65.58** | **56.92** | **22.44** |

### 4.5 ADDITIONAL ABLATIONS

**Choice of feature reconstruction.** mFRESA builds upon StableLSD but contains two new decoders, namely the feature and HOG decoders. In Table 6, we quantify the effect of each decoder. We observe that simply adding the DINOv2 feature decoder improves results across almost all tasks. Additionally, incorporating HOG features further enhances downstream performance, underscoring the effectiveness of both decoders in learning more accurate slot representations.

**Limitations.** Though our evaluation framework offers clear benefits over existing protocols, it has limitations. It is more computationally expensive than methods like linear or transformer probing (Mamaghan et al., 2025), but in return enables multi-axis evaluation of slot properties without training separate models for each dimension. Our study focuses on unsupervised OCL methods for images, yet the framework could be extended to video data (Elsayed et al., 2022; Kipf et al., 2022; Aydemir et al., 2023; Zadaianchuk et al., 2023) and weakly supervised approaches (Singh et al., 2025; Didolkar et al., 2025a). Finally, while our protocol relies on the enhanced grounded GQA dataset, additionally applying AwGA to datasets with grounding masks in novel settings (*e.g.*, underwater environments) would yield a more comprehensive benchmark.

## 5 CONCLUSION

Object-centric learning (OCL) has made notable progress in unsupervised object discovery (UOD) for real-world scenes. However, we find that UOD metrics fail to capture the true representational quality of object-centric latent representations. We propose a new evaluation protocol based on visual instruction tuning of a VLM that leverages the open-world reasoning abilities of large language models to assess broader OCL goals, including compositionality, OOD generalisation, and counterfactual reasoning. This overcomes limitations of existing evaluation schemes and allows to conduct multi-faceted analyses of OCL methods without expensive re-training. Moreover, we quantify issues with existing evaluation metrics and propose an attribution-aware grounded accuracy (AwGA) metric that jointly measures representation and localisation quality of OCL methods. We found that present OCL methods slightly lag behind foundational models on many VQA tasks. However, we show that this gap can be narrowed by combining complementary reconstruction heads within OCL. The resulting mFRESA surpasses state-of-the-art OCL methods across a variety of tasks.

## 6 REPRODUCIBILITY

We provide hyperparameters and training details for our VLM-based evaluation of the considered OCL methods in Sec. 4. The training details and network diagram for mFRESA are provided in Table 7. Together with the final paper, we will release the implementation and model checkpoints to reproduce the results.

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

Figure 7: **Multi-feature reconstruction for slot attention (mFRESA)** uses a DINOv2 (Oquab et al., 2024) model as a feature encoder network. The slot-attention module groups the obtained features into slots. Multiple decoders reconstruct the image, HOG features, and DINOv2 features from the slots. The slot-attention module, and the HOG and feature decoders are trainable, while DINOv2 and the image decoder model, a diffusion decoder, are kept frozen. The model is trained with Equation (3). Not visualised: The HOG features are computed according to (Dalal & Triggs, 2005).

# APPENDIX

## A    AN IMPROVED BASELINE

We here provide additional details of our proposed baseline, mFRESA, which builds upon the StableLSD framework (Jiang et al., 2023). StableLSD is an encoder-decoder architecture with a slot-attention bottleneck. It employs a DINOv2 model as the encoder, and a frozen Stable Diffusion (Rombach et al., 2022) model as the decoder. The slot attention module is trained using an image reconstruction loss. We extend this design by introducing two additional decoders: a HOG feature decoder and a DINOv2 feature decoder.

Given the slots, the *HOG decoder* reconstructs the HOG feature map (Dalal & Triggs, 2005) of the input image, encouraging slots to better capture object boundaries through edge information. HOG features are computed by aggregating gradient orientations within local neighbourhoods. The *DINOv2 feature decoder*, inspired by DINOSAUR (Seitzer et al., 2023), reconstructs DINOv2 features from the slots, complementing image-level supervision. The overall training objective is given as

$$\mathcal{L} = L_2(I_{\text{inp}}, I_{\text{recon}}) + L_2(F_{\text{inp}}, F_{\text{recon}}) + L_2(H_{\text{inp}}, H_{\text{recon}}), \tag{3}$$

where $I$ denotes images, $F$ DINOv2 features, and $H$ HOG features of the input and reconstruction, respectively.

The key contribution of mFRESA is the *joint reconstruction of image, feature, and edge signals*, enabling slots to learn stronger object-centric representations. A detailed network diagram is shown in Fig. 7.

mFRESA is trained on a single NVIDIA A100 GPU with 80GB of VRAM. The encoder and image decoder components closely follow the StableLSD setup (Jiang et al., 2023), with mFRESA introducing two additional modules: a HOG feature extractor (Dalal & Triggs, 2005; Wei et al., 2022) and decoder, as well as a DINOv2 feature decoder. The model is trained for 500K iterations on the COCO dataset (Lin et al., 2014). Images fed to the DINOv2 encoder (Oquab et al., 2024) are resized and centre-cropped to 518×518 pixels. We used an Adam optimiser (Kingma & Ba, 2014) for training our model for 500K iterations, similar to StableLSD. The full training and architectural details of our method are shown in Table 7. The additional decoders also lead to additional training times, for example, StableLSD on a single A100 GPU trains in approx. 43 hours. Adding the feature decoder increases the training time to 65 hours. Adding the HOG decoder further increases the total training time to 82 hours.

Table 7: **Architectural and training details for mFRESA.**

| Module | Hyperparameter | Value |
|---|---|---|
| General | Batch size | 32 |
| | Precision | fp16 |
| | Learning rate | 2e-5 |
| | Learning rate scheduler | Constant |
| | Optimizer | Adam (Kingma & Ba, 2014) |
| | Adam ($\beta_1$, $\beta_2$) | (0.9, 0.999) |
| | Adam eps | 1e-8 |
| | Weight decay | 1e-2 |
| | Learning rate scheduler | Constant |
| | Iterations | 500K |
| | Max. grad norm | 1.00 |
| Encoder | Architecture | DINOv2 (Oquab et al., 2024) |
| | Patch size | 14 |
| | Backbone | ViT-B (Dosovitskiy et al., 2021) |
| | Embedding dimensions | 768 |
| Slot Attention | # Iterations | 5 |
| | # Slots | 7 |
| | Slot Size | 768 |
| Image Decoder | Architecture | Stable Diffusion (Rombach et al., 2022) |
| | Model version | 2.1 |
| Feat. Decoder | Architecture | MLP |
| | No. of layers | 2 |
| | Hidden dimensions | 1536 |
| HOG Decoder | Architecture | MLP |
| | No. of layers | 2 |
| | Hidden dimensions | 1536 |

Table 8: **Type of properties encoded by slots**. VLM-based evaluation (accuracy in %, ↑) allows us to quantify the type of properties that a slot encodes via a categorization of the questions.

| Dataset | GQA (Hudson & Manning, 2019) | | | | MM-Vet (Yu et al., 2023) | |
|---|---|---|---|---|---|---|
| | Attribute | Category | Object | Relation | Recognition | Spatial |
| DINOSAURv2 (Seitzer et al., 2023) | 57.58 | 45.26 | 78.02 | 46.95 | 21.5 | 23.9 |
| FT DINOSAUR (Didolkar et al., 2025b) | 56.77 | 43.17 | **79.95** | 45.54 | 18.7 | 19.9 |
| SPOT (Kakogeorgiou et al., 2024) | 57.15 | 42.47 | 75.06 | 43.24 | 19.7 | 23.1 |
| Slot Diffusion (Wu et al., 2023b) | 57.40 | 39.77 | 73.52 | 41.28 | 20.5 | **28.9** |
| StableLSD (Jiang et al., 2023) | 56.73 | 43.43 | 75.19 | 44.20 | 21.6 | 22.3 |
| mFRESA *(ours)* | **59.08** | **46.74** | 77.63 | **47.23** | **21.9** | 22.1 |

## B ADDITIONAL RESULTS

### B.1 ADDITIONAL APPLICATION OF OUR EVALUATION FRAMEWORK

**Quantifying the type of learned slots.** Our evaluation framework can be used to to probe whether architectural choices bias slots toward encoding specific properties (*e.g.*, spatial, relational, or object). The GQA dataset (Hudson & Manning, 2019) categorises questions into four semantic types: *(1)* object (existence), *(2)* attribute (properties or position), *(3)* category (class membership), and *(4)* relation (subject–object relations). As shown in Table 8, feature reconstruction methods excel at existence and relation questions, whereas image-only methods like Slot Diffusion (Wu et al., 2023b) and StableLSD (Jiang et al., 2023) lag behind. mFRESA, which combines both, achieves the best results in three of four categories. For MM-Vet (Yu et al., 2023), covering recognition and spatial queries, results are mixed: feature- and image-based approaches perform similarly on recognition, while Slot Diffusion performs best on spatial relations.

**Correlation analysis.** We evaluate the robustness of our slot-attention evaluation framework to the choice of the large language model (LLM). While the choice of LLMs affects the absolute performance of the methods, we find that the relative ranking of OCL methods remains largely unchanged across different LLMs. Table 9 reports Spearman's rank correlations between results obtained with Phi2 and Qwen2-7B across multiple datasets. The high correlations ($\rho \geq 0.89$)

indicate that our proposed VLM-based evaluation framework is stable, with model rankings preserved regardless of the LLM used.

Table 9: **Spearman rank correlations between different models when using Phi2 and Qwen2-7B models as LLMs.** The results show that our evaluation framework is robust to the choice of LLM and the rank between the models, even with different LLMs, is largely preserved (very strong correlation).

| Dataset | GQA | POPE | MME | MMVet | VQAv2 | OOD | Nat. Bench | Sugar C. | AwGA |
|---|---|---|---|---|---|---|---|---|---|
| Spearman $\rho$ | 0.98 | 0.95 | 0.70 | 0.76 | 0.98 | 0.86 | 0.91 | 0.85 | 0.92 |

**Qualitative results.** Fig. 8 presents qualitative examples comparing the predicted masks with attribution-aware masks obtained by selecting the slots with the $K$-highest attributions based on a gradient-based attribution method (Simonyan et al., 2013). These results highlight that high-quality predicted masks or accuracy alone do not always imply the best slot representations. In some cases, the model produces an incorrect answer despite a seemingly accurate segmentation (*e.g.*, case *(ii)* SPOT), while in others the correct answer is derived from a distributed representation stored in a slot that does not correspond to the appropriate object (*e.g.*, cases *(ii)* and *(iii)*). This disconnect underscores that segmentation quality or predicted accuracy alone is insufficient for evaluating whether a slot captures the relevant reasoning signal.

**Mean and standard deviation of results.** Training the VLMs with different random seeds and evaluating the resulting models is computationally very expensive, as these VLMs are trained using 8 NVIDIA A100 GPUs, with the training time for the fine-tuning stage typically being around 24 hours. This makes it infeasible to provide the results for multiple runs via this approach. Instead, we report mean and standard deviation results for mFRESA and several baseline methods on representative datasets during *evaluation*. We set the temperature for the LLM generation to 0.02 and averaged the results over five random seeds (42, 1337, 2025, 4378, 8921). We report the results on SugarCrepe (Hsieh et al., 2023), MME (Fu et al., 2024), and POPE (Li et al., 2023) as representative datasets for visual question answering in Table 10. We use the accuracy as evaluation metric for the SugarCrepe and POPE datasets. For the MME dataset, we provide the score based on the MME evaluation script (with 2000 being the maximum for the perception task). Please note that the numbers reported in Table 1 and Table 2 of the main paper are for a temperature value set to 0. Comparing these to Table 10, we observe the ranking of the models following the same trend as with temperature 0. Setting the temperature $> 0$ allows to introduce randomness into the output of large language models, allowing us to obtain the mean and standard deviations during evaluation. Also note that mFRESA outperforms other methods on these datasets, even when measuring the results for multiple runs.

Table 10: **Mean and standard deviation of results.** Performance comparison with mean and standard deviation of different methods in a selection of representative datasets. SugarCrepe (Hsieh et al., 2023) and POPE (Li et al., 2023) are evaluated in terms of accuracy (in %, ↑), MME (Fu et al., 2024) in terms of its score (↑). We highlight the **best** and second best model among slot-attention methods.

| Method | SugarCrepe (Hsieh et al., 2023) | MME (Fu et al., 2024) | POPE (Li et al., 2023) |
|---|---|---|---|
| DINOv2 | 82.14 ± 0.13 | 1283.96 ± 07.29 | 82.08 ± 0.10 |
| DINOSAURv2 | 76.20 ± 0.25 | 1123.47 ± 12.63 | 81.84 ± 0.24 |
| FT-DINOSAUR | 71.25 ± 0.23 | 1016.15 ± 22.11 | 81.54 ± 0.18 |
| SPOT | 71.65 ± 0.15 | 1066.04 ± 07.19 | 79.69 ± 0.04 |
| Slot Diffusion | 70.23 ± 0.33 | 1090.75 ± 08.09 | 79.74 ± 0.12 |
| StableLSD | 72.89 ± 0.32 | 1126.06 ± 17.21 | 81.13 ± 0.11 |
| mFRESA *(ours)* | **77.18 ± 0.27** | **1184.40 ± 19.52** | **82.20 ± 0.08** |

## C DATASETS

Here we describe the datasets used in Sec. 4.1 for our VQA-based evaluation of OCL methods.

**VQAv2.0** (Goyal et al., 2017) is a dataset of 265,016 images from COCO and abstract scenes, each paired with an average of 5.4 open-ended questions requiring vision, language, and commonsense

reasoning. Each question includes 10 ground-truth answers and 3 plausible but likely incorrect ones, making it a robust benchmark for evaluating visual question answering (VQA) models.

**GQA** (Hudson & Manning, 2019) is a VQA dataset for real-world images that requires visual, spatial, and compositional reasoning. Importantly, GQA provides grounding masks (referred objects to answer questions) for each question for the validation set.

**POPE** (Li et al., 2023). The Polling-based Object Probing Evaluation (POPE) is designed to assess object-level perception and hallucination in vision-language models by querying the presence of specific objects in images. It consists of three settings: *(i)* Random – this setting samples absent objects at random, *(ii)* Popular – this setting selects missing objects from a frequently occurring object pool, and *(iii)* Adversarial – this setting targets commonly co-occurring but visually absent objects to challenge the model's grounding ability. In total, POPE consists of 3 sets of image-question pairs, each containing 1500 pairs with answer "Yes" and 1500 pairs with answer "No".

**MME** (Fu et al., 2024) is a comprehensive benchmark designed to evaluate the capabilities of multimodal large language models (MLLMs) across 14 diverse subtasks spanning both perception and cognition. In our work, we focus specifically on the perception tasks, which include coarse-grained recognition (existence, count, position, colour), fine-grained recognition (poster, celebrity, scene, landmark, artwork), and optical character recognition (OCR). Model performance on these tasks is measured using the perception score, capped at 2000 points.

**MM-Vet** (Yu et al., 2023). Unlike standard evaluation benchmarks, MM-Vet evaluates the integration of key vision-language (VL) capabilities, such as recognition, optical character recognition (OCR), knowledge reasoning, language generation, spatial understanding, and mathematical reasoning. MM-Vet contains 200 images and 218 questions, all paired with their respective ground truths.

## D  ENHANCED GROUNDED GQA DATASET

We construct our Enhanced Grounded GQA dataset in Sec. 4.4 based on the validation split of the original GQA dataset (Hudson & Manning, 2019). Our enhanced version comprises 10,000 questions, each accompanied by grounded segmentation masks. To convert grounding bounding boxes—i.e., the coordinates of objects referenced in the questions—into segmentation masks, we utilise the SAM2 model (Ravi et al., 2025), specifically the "sam2.1-heira-large" checkpoint with its default configuration.

To ensure relevance and clarity, we apply filtering criteria that discard images containing more than seven bounding boxes or where the total box coverage is less than 10% of the image area. These thresholds are chosen to retain only prominent objects while maintaining compatibility with object-centric learning (OCL) methods trained on the COCO dataset (Lin et al., 2014), which typically utilise seven slots. Additional examples from our Enhanced Grounded GQA dataset are shown in Fig. 9.

## E  USE OF LARGE LANGUAGE MODELS

During the preparation of this work, we used large language models (LLMs), specifically ChatGPT and Grammarly, to support the writing process. These tools were employed to improve clarity and readability by refining language and style. All substantive contributions, including the research idea, coding, analysis, and interpretation of results, were carried out solely by the authors.

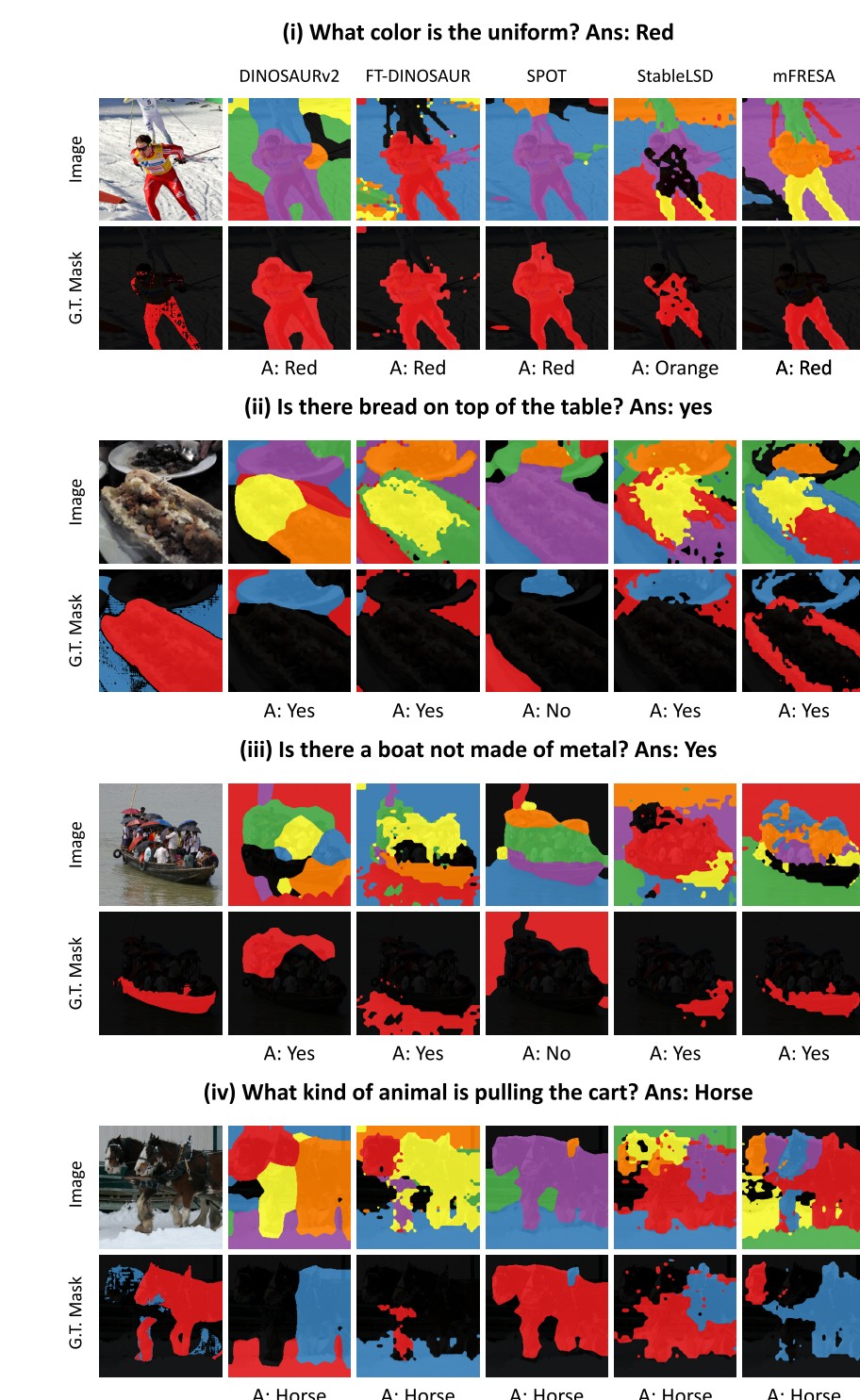

Figure 8: **Qualitative examples.** We visualize predicted masks and attribution-aware masks obtained by selecting the $K$ tokens with the highest attributions. A good predicted mask does not necessarily imply a correct slot encoding, which can lead to wrong answers (see *(ii)*, SPOT) or distributed representations where the correct answer is produced but derived using information stored in an incorrect slot (see *(ii)* and *(iii)*). As shown in *(i)* and *(ii)*, mFRESA built on StableLSD improves both segmentation and VQA results compared to StableLSD. G.T. Mask denotes the ground-truth GQA mask for the regions required to answer each question. In each example, the top rows show predicted masks; the bottom rows show attribution-aware masks.

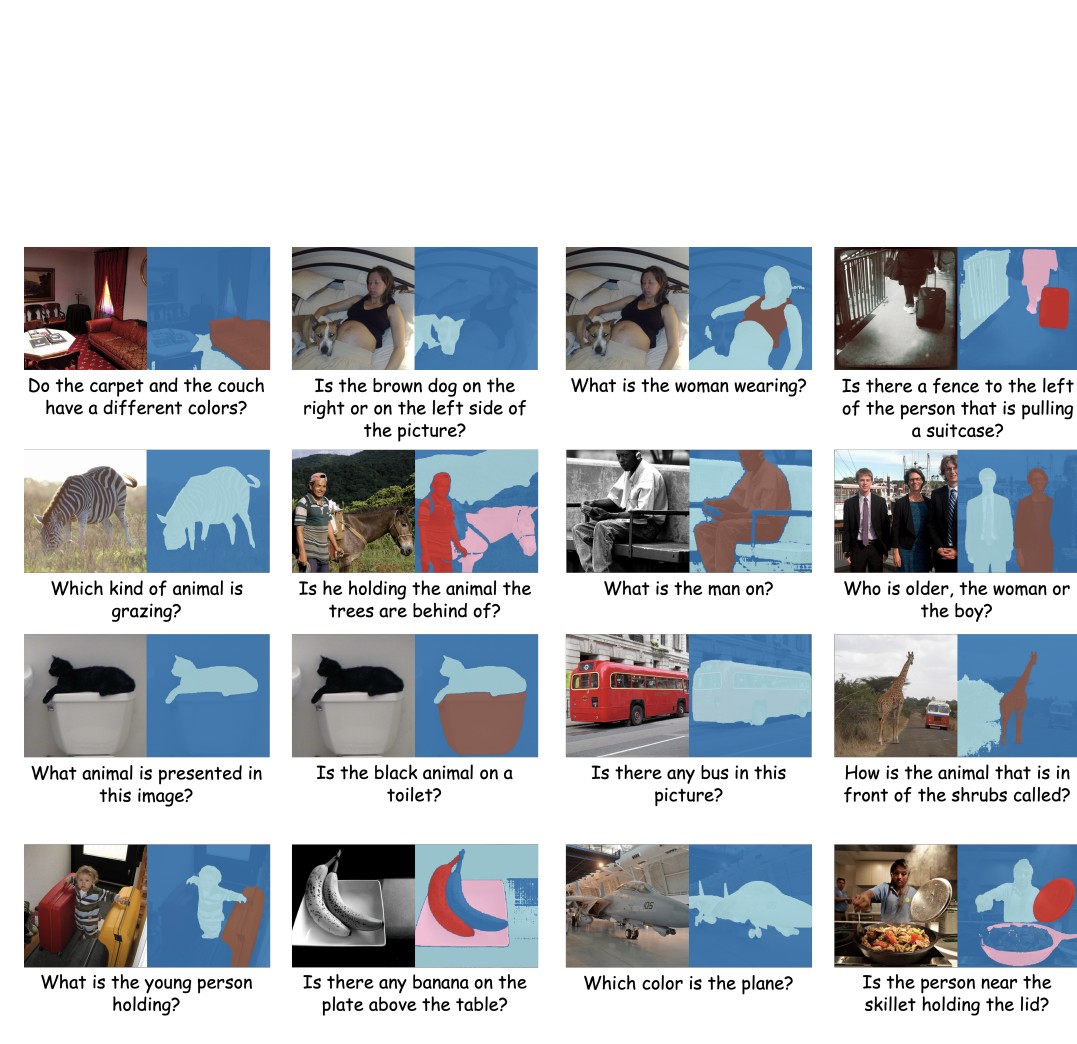

Figure 9: **Additional samples from the Enhanced Grounded GQA dataset.**

