# OpenReview forum: "Towards a more Holistic Evaluation of Object-Centric Learning"
_ICLR.cc/2026/Conference — Submitted to ICLR 2026_

### Official Review · Reviewer_9cBn · 2025-10-30

**Soundness:** 2
**Presentation:** 3
**Contribution:** 2
**Rating:** 4
**Confidence:** 3

**Summary:**

The paper introduces a new framework for evaluating object-centric learning (OCL) models beyond traditional object discovery tasks. It argues that existing metrics such as linear probing or segmentation accuracy fail to capture higher-level reasoning abilities like compositionality, out-of-distribution robustness, and counterfactual reasoning. More specifically, the authors highlight that current evaluation methods suffer from Type 1 inconsistencies (when a model predicts object properties correctly but localizes them poorly) and Type 2 inconsistencies (when multiple slots redundantly encode the same object, causing fragmented representations). To address these issues, the authors propose a Vision-Language Model (VLM)-based evaluation protocol, where OCL models act as vision encoders in visual question answering tasks, enabling multi-dimensional assessment. They also introduce a unified metric, Attribution-Aware Grounded Accuracy (AwGA), which jointly measures the “what” (object properties) and “where” (localization) aspects of object representations. Finally, they present mFRESA, a simple multi-feature reconstruction OCL baseline that outperforms existing methods across several holistic evaluation benchmarks.

**Strengths:**

1.	Addresses a critical gap in OCL evaluation.
The paper tackles the limitations of current evaluation schemes that focus mainly on object discovery and fail to assess higher-level reasoning such as compositionality, counterfactual reasoning, and out-of-distribution (OOD) generalization. It also clearly identifies Type 1 and Type 2 inconsistencies, motivating the need for a more holistic framework.

2.	Integration of VLMs as evaluators.
The authors propose using Vision-Language Models (VLMs) for evaluating OCL by treating OCL models as vision encoders connected to an LLM through a small MLP mapping. This design enables multi-dimensional assessment through visual question-answering tasks.

3.	Unified evaluation metric (AwGA).
The Attribution-Aware Grounded Accuracy (AwGA) metric effectively combines “what” (object properties) and “where” (localization) into a single measure and shows strong correlation with both, helping address the identified Type 1 and Type 2 issues.

4.	Clear and professional presentation.
The paper is clearly written and well-structured, with informative figures and detailed experimental descriptions that enhance readability and reproducibility.

5.	Practical baseline improvement (mFRESA).
The proposed mFRESA model, which uses multiple reconstruction targets, provides a simple yet effective improvement over existing OCL methods and supports the paper’s empirical claims.

**Weaknesses:**

1.	Evaluation may not isolate OCL representation quality (most critical).
Because the proposed framework embeds OCL encoders within a large Vision-Language Model (VLM), much of the reasoning can be performed by the language model itself. This makes it unclear whether the evaluation truly measures the representational quality of the OCL model or simply the VLM’s doing the heavy lifting. While linear probing is limited in scope, its simplicity ensures that it directly reflects representation quality; in contrast, using a complex multimodal pipeline introduces confounding factors that weaken interpretability.

2.	Limited analysis and validation of the AwGA metric (highly important).
The Attribution-Aware Grounded Accuracy (AwGA) metric depends on a Top-K attribution step that can still include redundant slots and may not fully resolve Type-2 inconsistencies. The paper provides no ablation or sensitivity study on K, leaving questions about how stable or fair the metric is across different settings.

3.	Heavy dependence on pretrained components.
Results may partly reflect the capabilities of large pretrained models (LLMs and feature encoders) rather than the OCL methods themselves.

4.	Overstated causal claims.
The evaluation benchmarks counterfactual question answering rather than true causal reasoning, so causal interpretability might remain unproven.

5.	High computational cost.
As mentioned in the paper, the proposed framework requires multi-stage fine-tuning of large models, which reduces scalability compared with simpler baselines like linear probes.

**Questions:**

1.	How do the authors ensure that the VLM-based framework measures OCL representation quality rather than the LLM’s reasoning ability?

2.	How sensitive is the AwGA metric to the choice of Top-K, and could redundancy among slots affect its reliability?

3.	Why is the proposed complex evaluation preferable to simpler linear probes that directly assess representation quality?

---

> ### Author Response · Authors · 2025-11-28
>
> **Q1, W1,W3 . How do the authors ensure that the VLM-based framework measures OCL representation quality rather than the LLM’s reasoning ability?**
>
> To verify that the reasoning ability comes from the OCL representation rather than the LLM alone, we disable the vision encoder by replacing all visual tokens with random embeddings. This analysis helps us to isolate the effect of the visual representations on performance across several benchmarks. As shown in Table 1, the Blind-VLM model (LLM only) performs substantially worse than when using an OCL or DINOv2 encoder, indicating that vision tokens are needed to solve these problems. An interesting exception is CVQA Boolean, where the blind model even outperforms DINOv2. These Boolean queries need little visual grounding. LLMs can often answer them from text alone because they follow simple templates (“Would X still be true if Y changed?”). Adding vision features can introduce noise and slightly hurt performance in such cases. In contrast, direct counterfactual questions (“How many X would there be if two X were added/removed?”) require inspecting the image, so visual grounding is essential, and models with vision tokens outperform Blind-VLM. We have added this explanation and also clarified claims about the performance of OCL methods on counterfactual question answering in *paragraph Robust perception evaluation in Sec. 4.1.*
>
> Additionally, we are not the first to use VLMs for evaluating visual representations. Cambrian-1 (Tong et al., NeurIPS 2024) (Ln 135-137) adopts a similar approach for models such as DINO and CLIP.
>
> **Q2. Sensitivity of the AwGA metric to the choice of Top-K.**
>
> K in Top-K choice is not a hyperparameter but a value that is set on a per-image/question pair basis. For example, if the number of grounding masks (objects required to answer the questions) is n, then K is set to n. We have clarified this further in the updated version (Ln 414-415).
>
> **Redundant slots in Top-K.** Redundancy in Top-K slots can occur, causing Type2 inconsistencies. This occurs, for example, when two slots (say A and B) encode parts of the same object (O1). Now, assume the grounding mask has two objects, say O1 and O2  (the objects required to answer the input question). Here, K would be 2 (number of grounding masks). Now, redundancy in Top-K slots can occur if the attributions for slots A and B are the highest. The AwGA metric would correctly penalize this, as there would be minimal intersection between the grounding mask of O2 and the masks of A and B. We also evaluated the effect of the attribution method on the AwGA metric (Table 5), however found the AwGA metrics to be quite robust to the choice of the attribution method.
>
> **Q3, W5. Preference of VLM vs. Linear Probes, higher computation cost**
>
> There are several reasons for preferring VLMs over linear probes for evaluating OCL models. Namely,
>
> 1. A more holistic evaluation of the representation power of an OCL method requires evaluation across multiple dimensions, e.g.,out-of-distribution generalisation, compositional generalisation, counterfactual reasoning, etc. However, scaling linear probing to multiple benchmarks would require retraining it repeatedly, which is cumbersome and expensive. For example, probing StableLSD on COCO takes 1.5–2 hours; scaling this to ten tasks becomes comparable in cost to our method. Although VLM training has a higher one-time cost, it supports zero-shot evaluation across diverse benchmarks, even potential new ones, something linear probes cannot do efficiently.
>
> 2. Additionally, testing certain properties, such as real-world counterfactual reasoning (“What would happen if…?”) is non-trivial with linear probing as it requires a discrete output space, which can grow easily very large, making optimization difficult. Alternatively, grouping the output space into classes is inherently ambiguous and would result in a coarse evaluation.
>
> 3. Also, prior works have already questioned the effectiveness of linear probes, motivating the use of stronger evaluators such as MLLM-based [Tong el al., NeurIPS 2024] and transformer-based [Mamaghan et al., ICLR 2025] probes—an approach our VLM-based benchmark follows.
> We have further strengthened this argumentation in the related work section, where we motivate the use of VLM-based probing models as explained above.
>
> **W4. Overstated causal claims.**
>
> Thank you for raising this point. You are correct that our evaluation focuses on counterfactual question answering rather than full causal reasoning. Our intention is not to claim that the model achieves true causal learning. Instead, our results show that the model can answer counterfactual queries, which may reflect learned patterns that mimic causal reasoning rather than genuine causal understanding. We have revised the text (Ln [301-303]) to clarify this distinction and ensure readers do not interpret counterfactual QA performance as evidence of full causal learning.

---

### Official Review · Reviewer_yKAi · 2025-10-31

**Soundness:** 2
**Presentation:** 3
**Contribution:** 3
**Rating:** 4
**Confidence:** 4

**Summary:**

This paper critiques the limited evaluation paradigms in object-centric learning (OCL), arguing that standard metrics like mean IoU or object discovery fail to assess what properties are actually represented in the learned slots. The authors propose a new evaluation framework using vision–language models (VLMs) built via visual instruction tuning (LLaVA-style) where object-centric encoders provide visual tokens to the LLM.
They introduce a new metric, Attribution-aware Grounded Accuracy (AwGA), which jointly measures “what” (object attributes) and “where” (localisation) by computing grounded accuracy weighted by slot-level attribution maps. They benchmark multiple OCL methods (e.g., DINOSAUR, FT-DINOSAUR, StableLSD, SPOT) on VQA, counterfactual reasoning, OOD generalisation, and compositional reasoning.
They also propose a new baseline, mFRESA, which adds multiple reconstruction targets (pixels, DINO features, HOG features) and shows improved performance on both VQA and AwGA metrics.

**Strengths:**

- The paper convincingly highlights a major gap in OCL evaluation: existing metrics are narrow and don’t reflect broader reasoning goals (OOD, compositionality, counterfactual reasoning).

- The distinction between Type1 and Type2 inconsistencies (localisation vs. redundancy) is well thought-out and nicely visualised.

- Novel metric (AwGA). A sound attempt to unify localisation and representation evaluation, addressing known shortcomings of mIoU and VQA-only accuracy.

- Evaluates a wide range of OCL methods across diverse tasks, providing a valuable comparative baseline for the field.

- Simple, interpretable baseline (mFRESA). The multi-feature reconstruction approach is practical and demonstrates that better feature-level reconstruction helps representation quality.

**Weaknesses:**

* From my understanding, the proposed framework evaluates through a language-mediated bottleneck, meaning performance is confounded by how the LLM integrates slots via cross-attention rather than by intrinsic slot quality.  For instance, two encoders with very different slot semantics could yield similar VQA results if the LLM adapts its attention weights effectively.
* Thus, the claim that this is a holistic evaluation is weakened by the absence of direct geometric or retrieval-based analyses (e.g., nearest-neighbour retrieval between slots, clustering consistency, object-level contrastive evaluation).
* There seems to be an overreliance on LLaVA-style architecture. The improvements might partly come from instruction tuning and alignment quality of the VLM, not necessarily from the OCL model itself. There’s limited analysis of how much the connector or instruction tuning dominates the results.

**Questions:**

1. How sensitive are the results to the choice of LLM and connector architecture?

2. Could AwGA be applied without ground-truth masks, e.g., via attention supervision or self-generated masks? Otherwise its scalability remains limited.

3. How much do you believe instruction-tuning data leakage affect fairness? Many LLaVA datasets contain COCO images, which overlap with OCL training data.

4. Would instance retrieval or clustering consistency across views yield the same model rankings as AwGA? This could validate whether AwGA captures true representational quality. The result will not be exact, but one would expect that a good representation captures both what and where of the object in the scene.

5. For mFRESA: do the additional reconstruction heads improve binding (object separation) or attribute encoding? A disentanglement or slot purity analysis would clarify this.

---

> ### Author Response · Authors · 2025-11-20
>
> We thank the reviewer for their comments.
> We would be grateful if the reviewer could elaborate on what they mean by "Would instance retrieval or clustering consistency across views yield the same model rankings as AwGA?".
> We are unsure what the reviewer exactly means and are not aware of any immediate literature that has evaluated object-centric learning methods in this manner. It would be really helpful if the reviewer could clarify this or point to the paper where this evaluation was used.

---

> > ### Comment · Reviewer_yKAi · 2025-11-26
> >
> > Sorry for the late response!
> >
> > I was referring to the quality of the standalone representations. My main concern is that the VQA setup on which mFresa is tested on, which is essentially a classification setup may be contributing to the method's performance. Thus a small instance retrieval experiment where you show that mFresa retrieves more faithful samples from its learned representations would show its effectiveness better.
> >
> > Please do let me know iif mFresa is not confounded with VQA objective and my interpretation is wrong.

---

> ### Author Response · Authors · 2025-11-28
>
> **W1, W3, Q1. Sensitivity to LLM and connector choices**
>
> We agree with the reviewer that this is an important point; therefore, we had already evaluated multiple LLM and connector choices. The results were presented in **Figs. 5 and 6, as well as in Table 9, and in Sec. 4.4**, paragraph *"Robustness of AwGA to LLM and Connector Choice."* Furthermore, **Tables 1, 2, and 4** also show the impact of using different LLMs.
> We found that our evaluation protocol is robust to the LLM and connector choices, showing a **ranked correlation of 0.93 for different LLMs and 0.71 for different connectors**. These ranked correlation values are considered very strong or strong, indicating the robustness of our evaluation protocol.
>
> **Please also see our updated Tables 1 and 2**. We have now included results for the BlindVLM model, a model with the vision encoder disabled (replacing all visual tokens with random values). This analysis helps us to isolate the effect of the visual representations on performance across several benchmarks
>
> **Q2. Scalability of the AwGA metric**
>
> In theory, AwGA can also be applied to self-generated masks. However, we would like to point out that when applying AwGA with self-generated masks, all methods should be evaluated using the same masks for a fair comparison. Moreover, since our work aims to benchmark current object-centric learning methods, it is most sensible to use highly accurate masks for the most robust ranking of these methods.
>
> **Q3. Data leakage issue**
>
> We thank the reviewer for this in-depth comment and analysis. **Regarding fairness, we believe that any data leakage has a limited impact when comparing different OCL methods, since the relative ranking between these methods is unaffected—all models are trained on the same dataset (COCO train split)**. Moreover, these models are trained in an unsupervised manner and never see labels for the VQA tasks. Additionally, we explicitly evaluated generalization on out-of-distribution samples and reported these results in Table 2 (OODCV dataset). Only DINOv2 is trained on a significantly larger dataset; however, it is only included as a reference upper bound of performance achieved by self-supervised approaches (Ln 219). If the reviewer still feels clarification is needed for the readers, we are happy to further emphasize this in the final version.
>
> **W2,Q4. Instance retrieval**
>
> **mFRESA is not confounded with VQA performance**. mFRESA is trained using the standard losses (reconstruction of self-supervised signals) already used in OCL methods. mFRESA, like other OCL methods, is a fully unsupervised approach. Thus, we believe there is no extra information that mFRESA uses that can cause confounding with VQA performance compared to other OCL methods.
>
> **Q5. Does mFRESA improve binding or attribute encoding?**
>
> The mIoU metric is suitable for assessing the binding power of OCL methods. As shown in **Table 4**, mFRESA exhibits slightly lower binding power (lower mIoU) compared to FT-DINOSAUR on our Enhanced Grounded GQA dataset. However, mFRESA demonstrates significantly better representation power, as shown by the accuracy metric, where it outperforms all other OCL methods. This indicates that **mFRESA improves the representation quality more**. Further, mFRESA improves both binding and attribute encoding performance in comparison to StableLSD, on which it is built (**Table 6**).

---

### Official Review · Reviewer_DkLU · 2025-10-31

**Soundness:** 3
**Presentation:** 4
**Contribution:** 3
**Rating:** 6
**Confidence:** 5

**Summary:**

The paper proposes a broader evaluation protocol for object-centric learning (OCL), arguing that standard object discovery metrics (e.g., segmentation quality) do not reflect downstream reasoning and robustness. It evaluates slot-based models by integrating them into a vision-language pipeline and testing them on diverse VQA-style, compositional, OOD, and counterfactual benchmarks. The authors also introduce AwGA, a metric that jointly measures answer correctness and whether the model used the correct object regions. They further present mFRESA, a multi-feature reconstruction variant of an object-centric model, which they claim improves both grounding and downstream robustness compared to prior slot-based methods.

**Strengths:**

1. The paper is well written, clear, and easy to follow.
2. The VLM-based evaluation pipeline for OC models enables a wide range of evaluations across perception, reasoning, robustness, and compositionality.

**Weaknesses:**

1. Some of the claims have already appeared in the literature, although under different experimental setups. Please see the questions below for more details.

**Questions:**

1. As my main concern: several claims and takeaways in the paper seem to have already been shown in prior work, but with a different evaluation pipeline. For example, [1] has already analyzed (1) comparisons between foundation models and OC models in terms of raw VQA performance, and (2) the correlation between unsupervised object discovery and downstream reasoning performance. In addition, [2] analyzes compositional generalization of OC models compared to foundation models in a fully controlled setting, showing advantages for OC models. Given this, it seems that the main contributions of the current paper go down to (1) the VLM-based evaluation pipeline, (2) the introduction of mFRESA, and (3) the AwGA metric. I would appreciate it if the authors could elaborate on this.
2. As an addition, I would like to see an ablation on the loss term weights for mFRESA to see how much each reconstruction target (e.g. pixels, DINOv2 features, HOG features) affects the downstream/upstream performance of the model.
3. To further enhance the results of the paper, I would also recommend showing qualitative effectiveness of mFRESA compared to other methods e.g. showing attribution maps for a few questions or (if possible) attention maps over slots from the LLM, to show which slots are actually being used for question answering.
4. In Section 4.1 (last paragraph), it is mentioned that on the compositional reasoning task of SugarCrepe, OC models lag behind foundation models like DINOv2. On the other hand, on similar (but synthetic) compositional generalization tasks, [2] shows that OC representations generalize better compositionally. Could you please elaborate on the differences between these two papers and why the trends seem reversed?
5. Minor typo: Line 147 defines S as a k-slot vector, but the indices are listed from 0 to k.

Given all the above, I believe this paper is taking an important and necessary step towards better understanding the role of object-centric learning in the current era of foundation models, and I would recommend the acceptance of the paper.

---

> ### Author Response · Authors · 2025-11-16
>
> We thank the reviewer for their comments. The reviewer probably forgot to include the citations [1] and [2] in the review. Could you please provide the paper citations so that we can address the concerns?

---

> > ### Comment · Reviewer_DkLU · 2025-11-17
> >
> > Yes, I am sorry about that. Please find the references below:
> >
> > [1] Mamaghan, Amir Mohammad Karimi, et al. "Exploring the Effectiveness of Object-Centric Representations in Visual Question Answering: Comparative Insights with Foundation Models." The Thirteenth International Conference on Learning Representations. 2025.
> >
> > [2] Kapl, Ferdinand, et al. "Object-centric representations generalize better compositionally with less compute." _ICLR 2025 Workshop on World Models: Understanding, Modelling and Scaling_. 2025.

---

> ### Author Response · Authors · 2025-11-28
>
> **Q1. Prior works, such as [1] and [2], have already evaluated some of the criteria.**
>
> We agree, and we discuss the limitations of transformer-based probing in [1] in Ln [125-129]. Briefly, [1] trains a transformer classifier directly on VQAv2 and GQA, thereby restricting evaluation to these two datasets. In contrast, our VLM-probe is trained on a much larger and more diverse corpus, enabling zero-shot evaluation of OCL methods on multiple dimensions (compositional reasoning, robustness to natural adversarial examples, etc.) that are beyond those included in VQAv2 or GQA benchmarks. Moreover, [1] relies solely on accuracy, which makes it susceptible to Type2 inconsistencies (i.e., fragmented object representations across slots). Our use of AwGA directly addresses this issue.
>
> [2] extends [1] but focuses exclusively on compositional generalization in synthetic environments such as ClevrTex. By comparison, our VLM-based benchmark trains a single model and reuses it across a wide range of evaluations, including OOD generalization, compositional reasoning, and adversarial robustness. Since [2] inherits the evaluation protocol of [1], it also inherits its sensitivity to Type2 inconsistencies. We have added citation [2] and explained how it differs from our work in Ln [125-128].
>
> In summary, beyond the contributions (1)–(3) noted by the reviewer, a key benefit of our approach is that it provides a single, unified evaluation framework, which, once trained, can assess multiple properties of OCL representations in a zero-shot manner.
>
> **Q2. Ablation for loss terms in mFRESA**
>
> Table 7 in the appendix contained such ablations. We have now moved this to the main paper (now Table 6).  Previously, this table only showed results for VQA tasks. We have now also added results showing the effect of adding the decoders on the AwGA and mIoU metrics. These results clearly show that adding the feature and HOG decoders helps improve performance on VQA tasks, as well as AwGA and mIoU metrics.
>
>
> **Q4. Reason for the difference with [2].**
>
> There are several differences between the evaluation setup in [2] and our method. (1) Unlike our analysis, which uses frozen OCL and DINOv2 models, [2] trains the OCL model directly on the CLEVRTex dataset, while the DINOv2 model remains untrained. Because DINOv2 is trained on natural images, the distribution gap between synthetic and natural images is substantial, which we believe contributes to its weaker performance in their setting. (2) [2] evaluates OCL methods on the compositional out-of-distribution (cOOD) split of the CLEVRTex dataset (unseen combinations within the same domain). This evaluation, however, entangles compositional generalization with OOD generalization. In contrast, our analysis explicitly disentangles these two aspects. As shown in Table 2, in the OOD setting, OCL methods are much closer to DINOv2. Taken together, these differences explain the divergence between our results and those reported in [2]. If the reviewer believes this explanation would be helpful to a reader, we would be happy to include it in the appendix.
>
> **Q3 & Q5** Thanks for the suggestion. We have added the suggested visualization (Fig. 8) and corrected the typo.

---

### Official Review · Reviewer_9mbB · 2025-11-02

**Soundness:** 3
**Presentation:** 3
**Contribution:** 3
**Rating:** 6
**Confidence:** 3

**Summary:**

The paper presents a new evaluation framework for object-centric learning using vision-language models. The idea is interesting and addresses a real gap in current OCL evaluation methods. The proposed metric and experiments are convincing. But the paper did no include necessary computation requirement data. Overall, it is a solid and meaningful contribution.

**Strengths:**

- originality: 3/5,
- quality: 3/5,
- clarity: 3/5,
- significance: 4/5. Four takeways are very valuable and worth-thinking for all future OCL researches.

**Weaknesses:**

W1
---
Line 073-076:
> Specifically, we employ the visual instruction tuning method of Liu et al. (2023), which modifies an LLM into a vision-language model (VLM). We use object centric models as the vision encoders, enabling us to evaluate OCL methods through visual question answering (VQA) via the VLM.

The proposed method is valuable and effective. However, the visual intruction tuning required in the authors' evaluation framework poses a very high computation demands, both in space and time. This even makes it impractical for most researchers, who have limited computation resources.

### W1.1

Figure 2:
Although stage 1 visual intruction tuning only requires tuning of "MLP Connector", the inference of LLM is still very expensive. Let along stage 2 needs tuning the LLM.

There fore, it is necessary to provide detailed training-time space and time costs, and the GPU numbers and models.



W2
---
Figure 7.
> Multi-feature reconstruction for slot attention (mFRESA).

The authors claim to propose a novel learning/optimizing method for OCL models. However, the reconstruction of VFM (vision foundation model) feature and the reconstruction of input pixel, except HOG, have already been used in many existing OCL methods. Besides, the two extra replica decoding times the training costs. Thus, the related data should also be provided.



W3
---
Section 2 Related Work:
Some important OCL advances are missing. Discussions should be provided on these OCL baselines and the ones being chosen in the experiment.
- SOLV: Self-supervised Object-Centric Learning for Videos
- VQ-VFM-OCL: Vector-Quantized Vision Foundation Models for Object-Centric Learning
- MetaSlot: MetaSlot: Break Through the Fixed Number of Slots in Object-Centric Learning
- VideoSAUR: Object-Centric Learning for Real-World Videos by Predicting Temporal Feature Similarities
- SlotContrast: Temporally Consistent Object-Centric Learning by Contrasting Slots

### W3.1

I am very interesting in the effect of slot pruning techniques, like SOLV and MetaSlot, on the authors' experiment results.

### W3.2

OCL can be conducted on both images and videos. Thus, it is suggested to also provide some results on video-based vision-lanaguage tasks. The corresponding OCL methods can be SOLV, VideoSAUR or SlotContrast.



W4
---
According to Table 6, only seven slots are used, which is very much less than the dense feature used in original VLMs. So it would make this work more complete if results of larger #slots are used, e.g., 16, 32 and 64.

**Questions:**

N.A.

---

> ### Author Response · Authors · 2025-11-28
>
> **W1**
>
> We appreciate the reviewer’s concern. Our VLM training utilized 8 × A100–80GB GPUs. Training times vary based on architecture and implementation choices. For example, for bulky models like StableLSD, pretraining takes about 4 hours, and finetuning takes 20–24 hours. We have added these details in Ln [258-261]. Training can also be done with fewer GPUs by increasing gradient accumulation. Importantly, once the VLM is trained, it enables zero-shot evaluation across multiple datasets and metrics, thereby amortizing the effective cost over all subsequent evaluations. Furthermore, linear probing for the property prediction task on the COCO dataset for StableLSD takes approximately 2 hours. The total time taken for learning different linear probes for evaluating different properties and the time taken to learn a single VLM-based probe are roughly comparable. Finally, we believe our analysis of different properties (Tables 2 and 3) makes a valuable contribution, despite the large evaluation times, as it sheds new light on the properties that existing OCL representations encode.
>
> **W2**
>
> The combination of all three reconstruction targets is a novel combination resulting in an improved baseline model compared to existing OCL methods. The training time on a single A100 GPU with 80GB vRAM for these models is the following: Img Only: Approx. 43 Hours (22GB vRAM), Img+Feat: 65 Hours (32GB vRAM), mFRESA: 82 Hours (36GB vRAM). We have added these details in Ln [807-809].
>
> **W3**
>
> We have included the suggested citations in Ln [504-505]. As our work focuses on image-based OCL methods, we did not include video-based methods such as Videosaur, Contrasting Slots, or SOLV in our experiments.  Given that VQ-VFM is an image-based model, we welcome the reviewer's suggestion and have included the results for VQ-VFM with Phi-2 LLM in Tables 1, 2, and 4. We are currently running the VQ-VFM model with Qwen2 LLM; we will incorporate its results once the training run finishes.
>
> **W3.1**
>
> Adaptive number of slots. MetaSlot code was released on September 10, approximately 15 days before the deadline, which made inclusion difficult. We attempted to use MetaSlot; however, no pre-trained models are available for direct evaluation. Once we successfully retrain their models from scratch, we will include these results as well.
>
> **W3.2**
>
> We thank the reviewer for these comments; however, the main focus of our work is to benchmark image-based OCL methods. Video methods are trained with video data, and to correctly evaluate these methods using VLMs would require VQA questions for videos. Further, comparing OCL methods trained with video data to image-based OCL methods trained with image data on COCO would lead to an unfair comparison, and we believe this muddies or confuses the main message of the paper for the reader. We note that follow-up work can perform similar benchmarking for video methods. However, this currently exceeds the scope of this work; we already discussed adding video models as future work in Ln [504]
>
> **W4**
>
> Since almost all existing OCL methods use 7 slots on COCO, increasing mFRESA’s slots would make the comparison inconsistent. Our goal is to maintain alignment between evaluations and existing OCL practices while utilizing VLMs as a unified benchmark for evaluation. To properly conduct the effect of slot numbers would require retraining all OCL methods with a higher number of slots, which is not possible during the given time limit. We will consider adding this analysis to the camera-ready.

---

### Author Response · Authors · 2025-11-28

We thank all Reviewers for their time and thoughtful feedback. We have provided detailed responses to each comment, clarified all raised points, and revised the manuscript accordingly. All modifications in the revised version are highlighted for ease of review.
Please let us know if any further clarification would be helpful.

---

### Author Response · Authors · 2025-12-03
**Message to AC (1/2)**

**To the Area Chair,**

We appreciate your effort and dedication during this challenging time. To support your decision process, we summarize the reviewers’ concerns and our responses. The most confident reviewer gave a rating of 6, and another reviewer also rated our paper as 6, with concerns primarily about computation time and the absence of video baselines. The two reviewers with ratings of 4 focused on isolating OCL representation quality and requested additional ablations. We highlighted our extensive LLM/connector robustness studies, added Blind-VLM results showing that OCL representations yield crucial benefits over LLM-only reasoning across benchmarks. We also noted that most AwGA ablations were already included but likely overlooked. Finally, we clarified the remaining minor issues. Below, we provide a detailed summary.

---

> ### Author Response · Authors · 2025-12-03
> **Message to AC (2/2)**
>
> ### **Reviewer 9cBn (Rating 4, Confidence 3\)**
>
> The reviewer agreed that the paper is well written, fills an important evaluation gap, and that AwGA, the VLM benchmark, and mFRESA are effective. Their main concerns were isolating LLM vs. OCL contributions, AwGA’s Top-K choice, the use of VLMs over linear probes, and causal phrasing.
> |Concern | Resolution |
> | :---- | :---- |
> | Isolation of LLM reasoning vs. OCL representation | Added Blind-VLM experiments (Tabs. 1–2) showing a clear drop without visual tokens, isolating and highlighting the contribution of OCL features. |
> | VLMs vs. linear probes | Showed that linear probing all properties is comparable in computational cost to our VLM evaluation, and that linear probes cannot handle tasks like counterfactual reasoning (Ln 119–124) and do not offer zero-shot evaluation across diverse benchmarks, including potentially new ones.  |
> | Sensitivity of AwGA to K (redundant slots) | Clarified and added that K is fixed by design (K is not a hyperparameter but equals the number of masks in the grounding mask, Ln 414-415); AwGA is robust across attribution methods (Tab. 5\) and penalizes redundant slots. |
> | Overstated causal claims | Clarified counterfactual QA ≠ causal reasoning in the text, updated Section 4.1 to further clarify that in the paper. |
>
> ---
>
> **Conclusion:** All concerns were addressed by adding new experiments, referencing existing ablations, or providing clarifications to the reviewer and incorporating them into the paper.
>
>
> ### **Reviewer yKAi (Rating 4, Confidence 4\)**
>
> The reviewer agreed that our work highlights a major gap in OCL evaluation, clearly identifies issues in existing metrics, provides a valuable comparative baseline, and that mFRESA is a simple, interpretable model that improves representation quality. The reviewer’s main concern focused on the sensitivity of our evaluation framework. Additional questions concerned scalability, potential data leakage, mFRESA’s performance characteristics, and distinctions between AwGA and VQA.
>
> |Concern | Resolution |
> | :---- | :---- |
> | Sensitivity to LLMs/connectors | Pointed to our existing robustness studies across LLMs and connectors (Figs. 5–6, Tab. 9). |
> | Scalability of the AwGA | Clarified AwGA can use self-generated masks if shared across methods; we used ground-truth masks for the most robust benchmarking. |
> | Data leakage from COCO | All OCL models are unsupervised and use the same data for training, thus rankings are fair; OOD results on OODCV (Tab. 2\) further mitigate this concern.  |
> | Confounding mFRESA and VQA | Clarified mFRESA is fully unsupervised, improving representation quality over binding (Table 4\) while enhancing both metrics compared to StableLSD. |
> | mFRESA improves binding or attr. encoding | Explained that mFRESA improves representation quality more than binding (Tab. 4), while still improving both compared to StableLSD (Tab. 6). |
>
> ---
>
> **Conclusion:** Most concerns stemmed from the overlooking of existing results; remaining issues were clarified.
>
>
> ### **Reviewer DkLU (Rating 6, Confidence 5\)**
>
> The reviewer praised the clarity and broad evaluation, raising minor concerns about differences to prior work, missing ablations, and missing visualizations.
>
> |Concern | Resolution |
> | :---- | :---- |
> | Differences from prior work | Clarified limitations of prior work compared to our method and why results differ from previous work (Kapl et al. 2025, Object-Centric Representations…, ICLR workshop) due to dataset and evaluation differences (Ln 125–129). |
> | Missing mFRESA ablations | Moved loss ablations into the main text (Tab. 6\) and added AwGA/mIoU results. |
> | Missing visualizations | Added requested visualization (Fig. 8\) and fixed minor typos. |
>
> ---
>
> **Conclusion:** All issues were minor and fully addressed.
>
>
> ### **Reviewer 9mbB (Rating 6, Confidence 3\)**
>
> The reviewer's main concerns focused on the computational cost and the addition of new baselines.
>
> |Concern | Resolution |
> | :---- | :---- |
> | Computation/time cost | Added full training-time and hardware details (Ln 258–261, 807–809); showed VLM cost amortizes and is comparable to the cost of multiple linear probes. |
> | Additional baselines | Added VQ-VFM (Tabs. 1, 2, 4\) as suggested; video-based methods deemed out of scope for image-based VQA benchmarking. |
> | Absence of adaptive slot methods, such as MetSlot | No pre-trained models and unstable public code; will be included once training issues are resolved after consulting the authors of MetaSlot. |
> | Increasing slot count | Increasing the number of slots impairs comparability (all prior methods use 7); retraining all models was not feasible. |
>
> ---
>
> **Conclusion:** All core issues have been resolved; remaining additional baseline requests can be added later without affecting the current conclusions or impact of work.

---

### Meta-Review · Area_Chair_p7po · 2025-12-28

**Summary:**

The paper initially received borderline recommendations with two weak accepts and two weak rejects. The primary concerns from the reviewers include: (1) concerns on the evaluation protocol, as the LLM/instruction-tuning pipeline may introduce additional confounds and may not effectively isolate the contribution of the visual representation itself, (2) concerns on novelty/contribution relative to recent works that study object-centric representations under similar VLM/VQA-style settings, (3) concerns on the AwGA metric, including sensitivity/scalability issues due to attribution choices and reliance on grounding masks, (4) practicality concerns due to the computational cost of the proposed protocol, and (5) concerns on missing baselines (e.g., adaptive-slot or video-based OCL).

**Reviewer Concerns:**

The rebuttal largely addresses the concerns regarding the Top-K selection in AwGA (by clarifying that K is set by the number of grounded objects rather than a tunable hyperparameter) and improves the positioning with respect to prior work. However, several important concerns remain outstanding or are only partially addressed, particularly those that directly affect the measurement validity of the proposed benchmark.

- **Does the VLM-based protocol truly isolate visual representation quality? (9cBn, yKAi)**

Unlike linear probing, the proposed evaluation trains a full multimodal system around each visual encoder. As a result, the final scores may reflect not only the information content of the visual representation, but also how easily the downstream model can *align to* and *exploit* that representation under the chosen training recipe (e.g., optimization dynamics and inductive biases of the connector/LLM). This makes it difficult to interpret the benchmark as a direct measure of *intrinsic* representation quality, since it can conflate representation quality with representation–LLM compatibility/alignability.

The added Blind-VLM experiment is a helpful sanity check demonstrating that visual tokens are necessary for many tasks. However, it does not resolve the key confound: it rules out the degenerate case where the LLM solves tasks purely from language priors, but it does not establish that *differences between encoders* are attributable primarily to the visual representations themselves rather than to differences in how the instruction-tuned model learns to use each encoder. In this sense, the protocol can be valuable as a measure of *MLLM system utility* (similar in spirit to work that compares visual features within an MLLM pipeline), but treating it as a direct proxy for representation quality is a stronger claim that requires additional validation.

- **Scalability and reliance on high-quality grounding masks in AwGA (9cBn, yKAi)**

The rebuttal clarifies the Top-K choice and notes that AwGA could, in principle, be applied with self-generated masks. Nevertheless, in its current form the metric still relies on accurate grounding masks that are shared across methods for fairness. Moreover, the “enhanced” masks are produced via a specific pipeline (e.g., converting boxes into masks), introducing an additional component whose quality may affect absolute scores and potentially influence relative rankings. It remains unclear how stable the rankings are when (1) masks are noisier, (2) masks come from different segmentation pipelines, or (3) ground-truth masks are unavailable. Thus, AwGA is currently most convincing under well-curated mask availability, while the broader claims about scalability and “holistic” applicability appear stronger than what is empirically demonstrated.

**Reviewer Scores:**

Given these remaining validity and scalability concerns, the AC believes the key objections from the negative reviewers would likely persist. Since this benchmark could influence evaluation practices in object-centric learning broadly, the AC recommends rejection this time.

---

### Decision · Program_Chairs · 2026-01-26

Reject